# The Fabrication of GaN Nanostructures Using Cost-Effective Methods for Application in Water Splitting

Xin Xi [1,2,*], Lixia Zhao [3,4], Tuo Li [1,2], Xiaodong Li [5] and Chao Yang [6]

1   State Key Laboratory of High-End Server & Storage Technology, Inspur Group Company Limited, Beijing 100084, China
2   Shandong Yunhai Guochuang Innovative Technology Co., Ltd., Jinan 271001, China
3   Institute of Semiconductor, Chinese Academy of Sciences, Beijing 100086, China
4   School of Electrical and Electronic Engineering, Tiangong University, Tianjin 300387, China
5   School of Electrical and Information Engineering, Wuxi University, Wuxi 214105, China
6   China Electronics Standardization Institute, Beijing 100176, China
*   Correspondence: xixin@semi.ac.cn; Tel.: +86-1861-4084-719

**Abstract:** The adjustable bandgap, single crystal structure, and strong chemical inertness of GaN materials make them excellent candidates for water splitting applications. The fabrication of GaN nanostructures can enhance their water splitting performance by increasing their surface area, improving photon absorption, and accelerating photocatalytic reactions. Developing cost-effective methods to fabricate GaN nanostructures is crucial to promote the development of GaN-based materials in water splitting applications. In this review, we introduce the main cost-effective techniques for the fabrication of GaN nanostructures and highlight future development directions.

**Keywords:** GaN; water splitting; nanostructures; nanoporous; nanocolumns





## 1. Introduction

The use of clean energy is the primary goal for the sustainable development of human society [1–3]. Conventional fossil fuels release contaminated residue and greenhouse gases, which cause serious harm to the environment [4–6]. As the environmental crisis intensifies, researching clean, safe, and sustainable energy sources is indispensable for replacing traditional fossil fuels. Water splitting is one of the most promising solutions for substitution, in which water is split into hydrogen and oxygen using solar energy. The hydrogen can be used as fuel, and the oxygen can be used in various chemical or biological applications. Throughout the whole process, no environmentally harmful products are released. Additionally, solar energy reserves are abundant, making energy exhaustion an unnecessary concern. Furthermore, water splitting has high combustion efficiency, enabling its use in vehicles, spacecraft propulsion, electric devices, and other applications [7–10].

In recent decades, many materials have been developed for water splitting applications, such as $TiO_2$ [11,12], ZnO [13,14], $BiVO_4$ [15,16], $Cu_2O$ [17,18], $WO_3$ [19,20], and so on [21–24]. However, most of the photocatalysts are polycrystalline in structure and suffer from reduced carrier transport efficiency due to scattering effects between grain boundaries. Furthermore, many photocatalysts have fragile chemical bonds [25–28] that cannot withstand harsh alkali or acid conditions and react with the electrolytes. Therefore, there is a pressing need to develop new photocatalysts with stable performance for water splitting applications.

The principle of the water splitting reaction mainly involves three stages, which are presented in Figure 1 [2–4]. The first stage is the generation of electron–hole pairs on the photoanode. When the energy of incident photons is greater than the bandgap of the photoanode, it excites many electron–hole pairs on its energy band. The second stage is the transfer of photo-generated electron–hole pairs to the surface of the photoanode. Due

to diffusion or polarization effects, the electron–hole pairs migrate toward the surface of the photoanode. During this process, some photo-generated carriers are trapped in crystal defects or recombine with each other, resulting in energy loss. Thus, improving the crystal quality and designing a fast separation mechanism for electron–hole pairs can significantly enhance the water splitting performance. The third stage involves the oxidation and reduction of $OH^-$ and $H^+$. If the electrons are more negative than 0 V vs. NHE (normal hydrogen electrode), they will reduce $H^+$ into $H_2$. If the holes are more positive than 1.23 V vs. NHE, they will oxidize $OH^-$ into $O_2$. The equation for the water splitting reaction is shown below:

$$H_2O \rightarrow 1/2O_2 + H_2; \Delta G = +237 \text{ kJ/mol}, E = 1.23 \text{ eV} \tag{1}$$

$$2H^+ + 2e^- \rightarrow H_2; (E_{Re} = 0 \text{ eV}) \tag{2}$$

$$2H_2O \rightarrow 4H^+ + 4e^- + O_2; (E_{Ox} = 1.23 \text{ eV}) \tag{3}$$

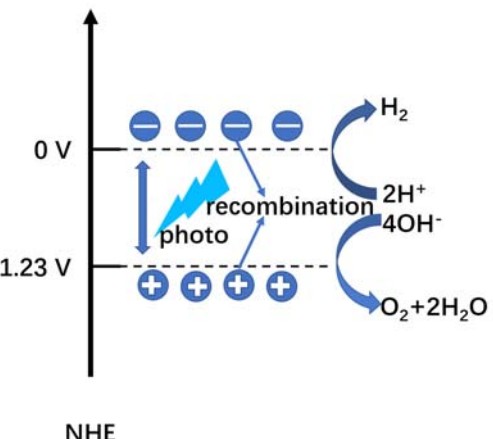

**Figure 1.** Schematic diagram of the water splitting process.

Therefore, the bandgap of the photoanode must be larger than 1.23 eV, with the potential of electrons on the conduction band more negative than 0 V and holes on the valence band more positive than 1.23 V vs. NHE. For fast separation of photo-generated carriers, the photoanode should be designed with a strong internal electric field to accelerate the transport of electron–hole pairs, a large surface-to-volume ratio to absorb more photons and provide more reaction sites, and a narrow bandgap to increase the solar absorption range.

As a third-generation semiconductor, GaN has been widely used in LEDs [29] (light-emitting diodes), LDs [30] (laser diodes), HEMTs [31] (high-electron-mobility transistors), and IGBTs [32] (insulated-gate bipolar transistors) due to its excellent optical-electronic properties. Typically grown as single crystals by MOCVD [33] (metal organic chemical vapor deposition), GaN exhibits superior carrier transport characteristics, with an electron saturation migration rate of up to $2.5 \times 10^7$ cm/s [34]. Meanwhile, GaN is a strong ionic crystal that is resistant to alkali or acid corrosion [35]. Moreover, GaN can form a ternary alloy with In, which can tune the bandgap from 0.7 to 3.4 eV [36]. The tunable bandgap enables GaN to absorb a wider range of the solar light spectrum, making it an ideal candidate for water splitting applications. Shyam first used GaN as a photoanode in 1995 to investigate the relation between the flat band and pH values [37]. Subsequently, Fujii systematically investigated the influence of various factors, including doping concentration, doping type, and polar facets, on the water splitting performance of GaN [38–41]. It was found that GaN with a higher doping concentration, n-doping, and polar facets exhibits better photocatalytic performance.

The fabrication of nanostructures is one of the most efficient methods to improve their water splitting performance [42–44]. It can not only increase specific areas but also enhance contact between the photoanode and electrolyte. Previous research has shown

that the fabrication of GaN nanostructures can effectively enhance their water splitting performance [42,45–48]. For example, S. W. Ryu fabricated GaN nanopores using an electrochemical etching method and found that GaN nanopores had better water splitting performance than planar GaN [43]. W. J. Tseng also prepared GaN nanopores and observed similar enhancements in water splitting performance [49]. Increased surface areas have been shown to enhance light absorption effectively [50–53]. Additionally, reduced sidewalls shorten the migration distance of photo-generated carriers to the GaN/solution interface, reducing the chance of electron–hole recombination [54,55]. Magnified surface areas introduce more surface states, creating an inner electric field that accelerates the separation of photo-generated carriers and hinders their recombination [52].

The fabrication methods of GaN nanostructures mainly include MOCVD, SAG, VLS, MBE, HVPE, electrochemical etching, and ICP etching [45,56–77]. Among them, MOCVD and SAG cannot fabricate a homogeneous morphology and stable properties of GaN nanostructures. MBE can grow materials with atom precision but is unsuitable for large-scale production due to its slow growth rate and limited growth area [62–64]. The HVPE method has difficulty in doping, which results in the inability to grow GaN-based materials with high carrier mobility.

The epitaxy technology for GaN wafers was developed from a prototype to commercial application over about thirty years. The epitaxy of GaN with a large size and high quality has been matured for mass production for a long time. Based on mass-produced GaN wafers, chemical etching can be adopted to fabricate GaN nanostructures with stable properties on a large scale. Therefore, the GaN nanostructures fabricated by chemical etching are currently the most cost-effective method for quantitative production.

Recently, a lot of research has been conducted on the development of GaN nanostructure fabrication via chemical etching. For instance, lateral nanoporous GaN etching technology enables the production of GaN nanopores in nanometer size. Moreover, an AAO mask can be used to fabricate uniform and ordered nanocolumns. All of these studies indicate that the preparation of nanostructures can significantly enhance the properties of photocatalytic water splitting. Building on these new findings, this review will focus on recent fabrication methods of GaN nanostructures using chemical etching. Additionally, we will discuss the application of GaN nanostructures in water splitting. Finally, we will highlight the most promising direction for GaN nanostructure fabrication.

## 2. Lateral Nanoporous GaN

Lateral nanoporous GaN structures are commonly fabricated by electrochemical etching [54,55,78–80]. The fabrication process is shown in Figure 2 [54]. Initially, a passive layer is deposited on GaN epilayers. Then, the sidewalls of the n-GaN epilayers are exposed using a laser scribing process. Subsequently, the GaN epilayers are immersed in the electrolyte and undergo the electrolysis process using a three-electrode cell.

As shown in Figure 3, the size and shape of the GaN nanopores exhibit different morphologies with changes in applied voltages. At higher voltages, the nanopores tend to form in a quasi-circular shape, while at low voltages they tend to form a triangular shape, as shown in Figure 3a. This indicates that GaN tends to undergo anisotropic electrochemical etching at lower voltages. The diameters of GaN nanopores increase with increasing applied voltage, as shown in Figure 3b. The formation mechanism of GaN nanopores is illustrated in Figure 2c. During the electrochemical etching process, 3 facets at (0001), (11–20), and (1–100) begin to undergo solution corrosion at 10 V. The facet (0001) possesses the lowest energy and is therefore preferentially corroded first, forming a triangular shape along the (0001) facet direction. As the applied voltage increases to 15 V, the corrosion speeds at facets of (1–100) and (11–20) begin to increase, resulting in an oval shape. Finally, when the applied voltage reaches 20 V, the corrosion speeds of all facets become almost the same, resulting in a quasi-circular shape.

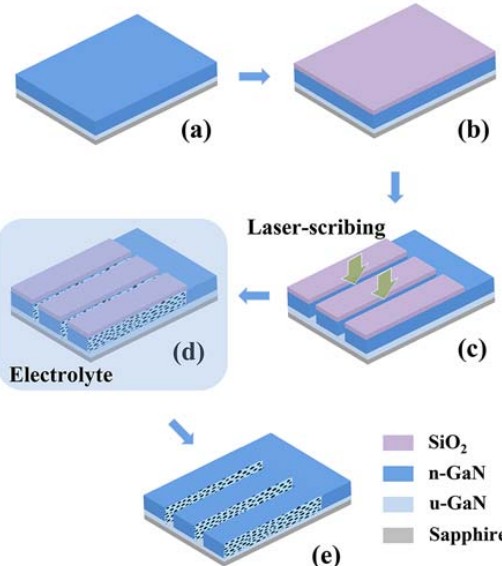

**Figure 2.** Schematic process diagram for the laterally porous GaN [54], as follows: (**a**) Epitaxial growth of GaN. (**b**) Deposition of SiO₂. (**c**) Laser scribing for the etching channels. (**d**) Anodic etching. (**e**) Removal of SiO₂. Reprinted with permission from [54] 2017, ACS.

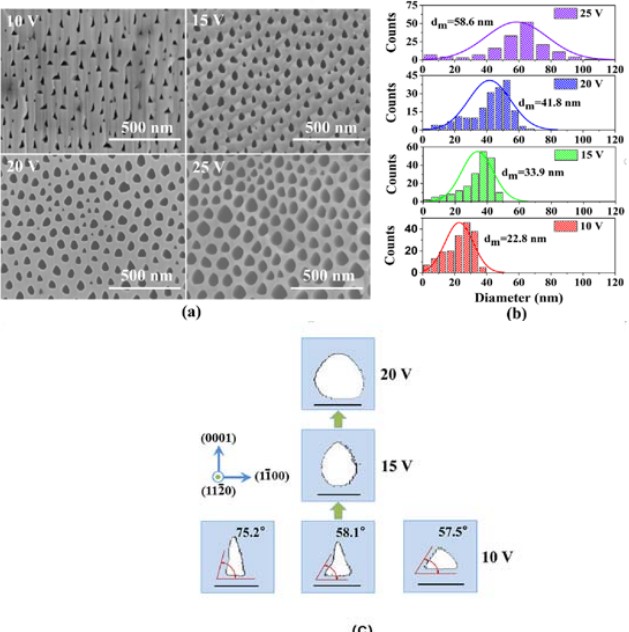

**Figure 3.** (**a**) Cross-sectional SEM images of the pore morphologies etched at different voltages [54]. (**b**) Statistics of the pore size distribution and the mean pore diameter. (**c**) Evolution of the typical pore morphologies with increasing voltage. The scale bar is 40 nm. Reprinted with permission from [54] 2017, ACS.

The water splitting performance of lateral GaN nanopores is better than that of planar GaN, as shown in Figure 4. Firstly, the surface area of nanoporous GaN is larger than that of planar GaN, which can reduce the current crowding at the GaN/solution interface and enhance water splitting efficiency. Secondly, the nanoporous GaN structure can shorten the carrier transport distance, effectively inhibiting carrier recombination. Additionally, the layer-ordered porous structure can enhance the light absorption through electromagnetic distribution, which, in turn, enhances the water splitting efficiency.

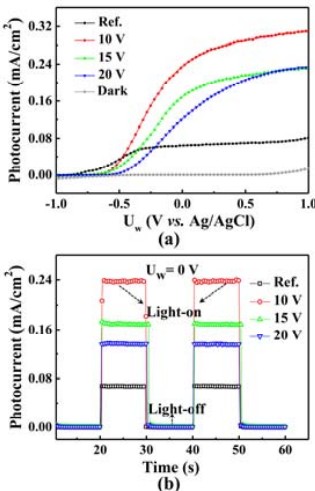

**Figure 4.** (**a**) J-V curves of the as-grown GaN and laterally porous GaN etched at different voltages [54]. (**b**) Time-dependent photocurrent density of different GaN photoanodes at 0 voltage with illumination period of 10 s. Reflectance of the planar GaN and laterally porous GaN etched at different voltages. Reprinted with permission from [54] 2017, ACS.

During the fabrication process of lateral GaN nanopores, regularly adjusting the applied voltage can yield various nanopore morphologies, including cone-shaped, bamboo-shaped, and tadpole-shaped, as shown in Figure 5 [79]. Asymmetrical sawtooth, symmetric sawtooth, square, and composite waveform anodization voltages are applied to obtain different pore morphologies. Pore size changes with the applied voltage directly. Diverse waveforms can even be combined to fabricate specific morphologies, such as cone–bamboo-shaped and others. Designing GaN nanoporous structures can form photonic crystal structures and enhance the light absorption for a certain range of wavelengths. FDTD simulation was used to analyze the electromagnetic distribution in different morphologies of GaN nanopores. The results demonstrate that the well-ordered periodic microcavities can modulate light distribution in three-dimensional directions by changing the shape and size of nanopores. That means that designing GaN nanopores with photonic crystal structures can confine light within the nanopores and enhance light absorption, thereby further improving water splitting performance.

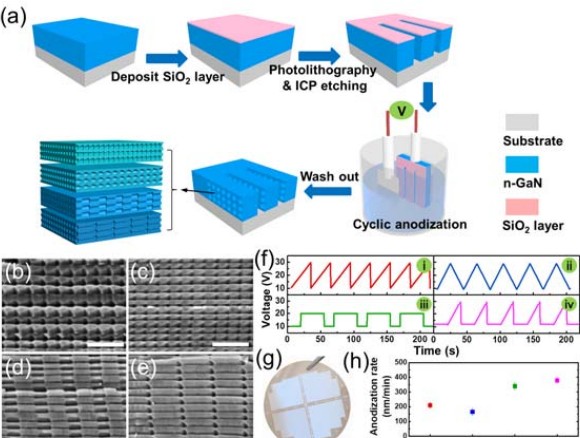

**Figure 5.** (**a**) The fabrication procedure for lateral periodic and shape-controllable nanoporous GaN. SEM images of (**b**) LN-GaN-C, (**c**) LN-GaN-E, (**d**) LN-GaN-B, and (**e**) LN-GaN-T. [80] Scale bar: 300 nm. (**f**) Typical cycles of applied voltage–time curves during cyclic anodization. (**g**) Digital photo of the 2-inch wafer LN-GaN-C. (**h**) Calculated anodization rate of different nanoporous GaN samples. Reprinted with permission from [79] 2020, Elsevier.

### 3. Vertical GaN Nanopores

Vertical nanoporous GaN can also be fabricated by electrochemical etching [48,52,81]. Figure 6 shows schematic diagrams of the formation process of vertical GaN nanopores under different voltages [81]. As the applied voltage increases, the size and porosity of the GaN nanopores become larger and denser. Mathematical statistics reveal the volume and density of the nanopore distribution shown in Figure 6g, which provides collaborative evidence of changes relative to the applied voltage.

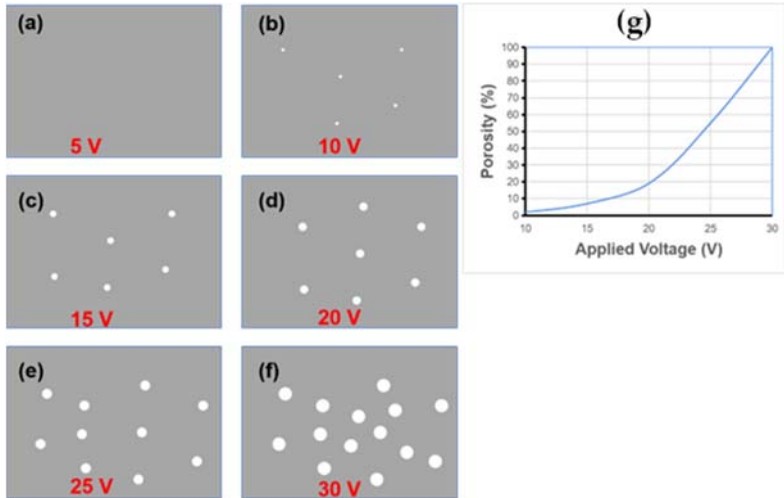

**Figure 6.** (**a–f**) Illustration of GaN nanopores after electrochemical etching under different voltages. (**g**) Porosity of GaN etched at various voltages; the porosity at 30 V is normalized as 100%.

Figure 7 shows the etching mechanism of porous GaN. The electrochemical etching process mainly consists of five stages. Stage 1 is pore initiation. When the voltage applied to planar GaN is high enough and the energy is sufficient to react with the electrolyte, the built-in electric field causes a large number of holes to be transported to the GaN/electrolyte interface. Defects on the GaN surface are more likely to trap carriers and react with the electrolyte, so the initial reaction sites are often located at defect sites, which are called initial sites. Stage 2 is pore branching. Due to the reaction at the initial sites, a large quantity of holes is generated by avalanche breakdown and accumulates at surface defect sites, resulting in the oxidation of GaN around the initial sites and multiple currents flowing around the center of the initial sites. Stage 3 is pore propagation. Multiple currents corrode GaN along the branches and form deep pores. In stage 4, the stable state, when the pore penetration approaches the bottom of GaN, the electrochemical etching evolves into the final stage, the stable stage. The GaN nanopores generated by the currents' deep penetration obtain high resistivity and decrease the electrochemical etching current, especially when the etching depth reaches the undoped GaN layer. As a result, the voltage drop at n-GaN does not have enough potential to sustain the pore etching. Throughout the process, a higher voltage significantly accelerates each stage and shortens the chemical etching period.

Numerous studies have shown that vertical nanoporous GaN enhances water splitting performance compared with planar GaN. The high porosity of nanoporous GaN increases the contact area with the electrolyte and provides more active areas for photocatalytic reactions. Additionally, nanoporous GaN can enhance light absorption through multiple reflections at the sidewalls of the nanopores. Furthermore, nanoporous GaN can shorten the transport distance for photo-generated carriers to the GaN/electrolyte interface. These advantages improve light absorption and reaction efficiency, leading to excellent water splitting performance.

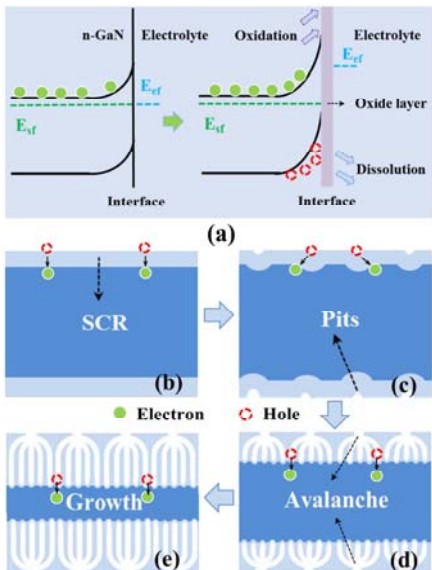

**Figure 7.** (**a**) Etching current during the entire electrochemical etching process. [54] (**b**) Defects at the surface of GaN. (**c**) Initial electrochemical etching pits. (**d**) The pore propagation process. (**e**) Stable stages. Reprinted with permission from [54] 2018, ACS.

To further increase the porosity of GaN nanopores, a combination of lateral and vertical nanopores has been developed, as shown in Figure 7 [56]. Based on the lateral GaN nanopore structure, vertical nanopores are etched by the ICP method using aluminum anode oxide (AAO) templates as a mask. After the etching process, the residual AAO template is removed by water flushing, resulting in a composite nanopore structure as shown in Figure 8d.

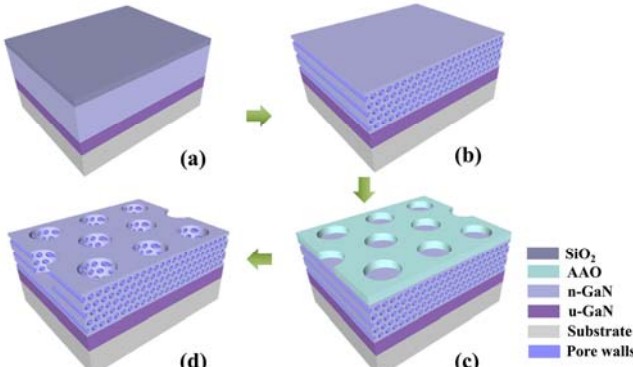

**Figure 8.** The schematic fabrication process for the composite porous GaN [55]. (**a**) Deposition of $SiO_2$ passivation layer. (**b**) Lateral anodic etching and removal of $SiO_2$. (**c**) Transfer of AAO membrane as a dry etching mask. (**d**) ICP etching and removal of AAO residues. Reprinted with permission from [55] 2018, ACS.

The SEM image of the composite GaN nanopore structure is shown in Figure 9c,d. As shown in the picture, the sidewalls of the GaN vertical nanopores are relatively smooth, indicating no significant etching damage caused by ICP etching. Figure 9c shows the angled-tilted (25°) SEM image of the composite nanoporous GaN, in which many lateral nanopores are embedded in the vertical nanopores. Due to the small diameters of the lateral nanopores of about 20~100 nm, the GaN nanopores have an extremely large length-to-diameter ratio, and gas has difficulty escaping from them in water splitting reactions. The vertical nanopores can promote the evolution of gas by channels at the sidewalls. Additionally, the composite nanopores can significantly magnify the surface-to-volume

ratio of GaN. Moreover, the inflow of electrolyte in the lateral nanopores is much easier for composite nanoporous GaN compared with lateral GaN. All of these improvements can significantly enhance the water splitting performance.

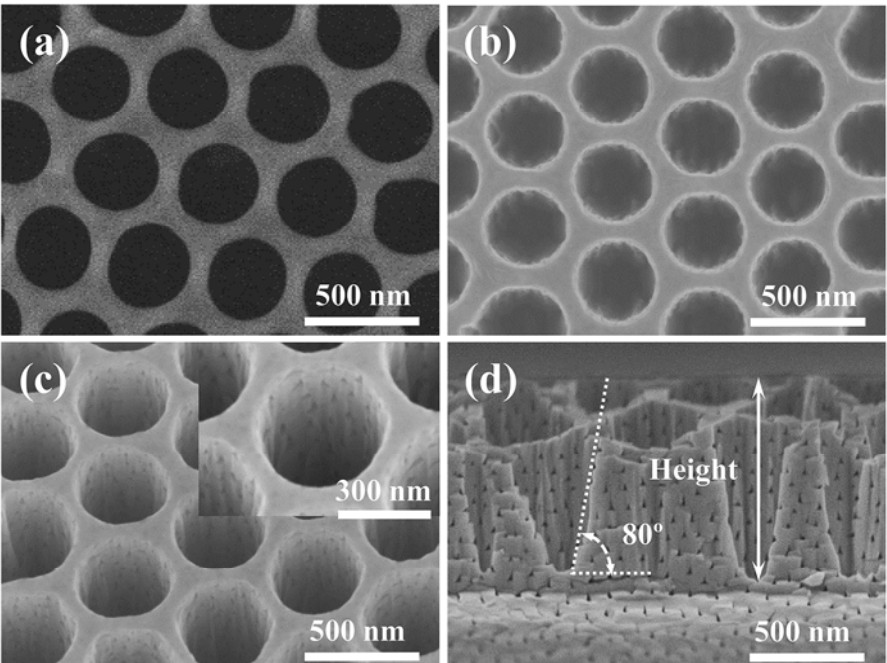

**Figure 9.** (**a**) Top-view SEM images of the AAO mask [55]. (**b**) Top-view SEM images of the composite porous GaN after ICP etching. (**c**,**d**) Angled-view (25°) and cross-section images of the composite porous GaN. The inset in (**c**) is a zoomed-in image of the structure unit of the vertical holes. Reprinted with permission from [55] 2018, ACS.

Besides the ventilation of gas and electrolyte to lateral nanopores, the composite nanoporous structure affects the reflectance of GaN. Figure 10a shows the reflectance of planar GaN, lateral nanoporous GaN, and composite nanoporous GaN. Due to the difference in refractive index between GaN and air, there is a divergence angle that restricts the emergence of light. For lateral nanoporous GaN, diverse pore shapes can increase light scattering inside GaN and boost the reflection of GaN. For composite nanoporous GaN, although the porous density increases, the reflection rate unexpectedly decreases. This mainly occurs due to multiple reflections between the internal adjacent sidewalls of vertical nanopores, trapping light in the vertical pores and absorbing it through the sidewalls repeatedly. Further evidence of enhanced light absorption was found in FDTD electric field calculations, which showed that ICP etching of vertical nanopores localizes the light by mutual reflection between adjacent nanopores.

The water splitting performances of planar GaN, lateral nanoporous GaN, and composite nanoporous GaN are shown in Figure 10b. The photocurrents increase in ascending order from planar GaN to composite nanoporous GaN. This is mainly due to the multiple influences of a larger surface area, higher optical absorption, and more convenient contact with the reaction substance. Figure 10c,d show the excellent photoelectrical response and stable anode working properties of composite nanoporous GaN.

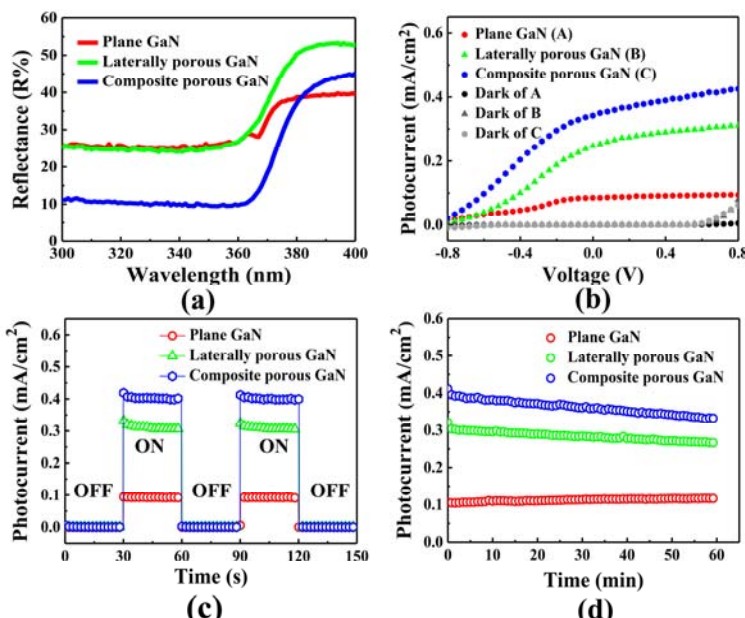

**Figure 10.** (**a**) Optical reflectance spectra of the planar GaN and porous GaN samples [55]. (**b**) J-V curves of the GaN photoelectrodes mentioned earlier. (**c**) I-t curves of different GaN photoelectrodes. (**d**) Stability of the photocurrents under illumination at 0.5 V for the 3 GaN photoelectrodes. Reprinted with permission from [55] 2018, ACS.

## 4. GaN Nanocolumns

For enhancing the water splitting performance, the fabrication of nanocolumns is another method of GaN nanostructure fabrication [44,82,83]. There are mainly two methods to fabricate GaN nanocolumns, including bottom-up and top-down technology. The bottom-up method mainly consists of material growth technology, such as SAG [73], MBE [82], VLS [66], etc. However, because of doping difficulties, unstable growth conditions, and inhomogeneity, it remains challenging to grow GaN nanocolumns using the bottom-up method in a cost-effective manner. Nevertheless, planar GaN growth and ICP etching technologies are mature, and GaN nanocolumns can be fabricated using a top-down etching method without difficulty [74,84]. In this technique, an etching mask is used to pattern the nanocolumn arrays that resist the ICP etching gas. Due to the selection ratio difference, the substrate under the mask is retained, whereas the exposed parts are etched. Thus, the fabrication of nanocolumn arrays is completed by transferring the pattern from the mask. Figure 11 shows the fabrication of GaN nanocolumns using AAO as an etching mask [84]. Initially, $SiO_2$ is deposited on GaN wafer as the hard mask to transfer the pattern from AAO thin film. Then, the AAO with well-arranged nanopores is transferred onto the $SiO_2$ layer. For a hard mask with good inertness to ICP etching, a Ni layer is deposited on an AAO membrane. The AAO layer is then removed with a fine arrangement of Ni particles. Using the Ni particles as a mask, the $SiO_2$ and GaN are both etched using the ICP method. Finally, the GaN nanocolumns are fabricated after ICP etching, as shown in Figure 12.

The PL and reflection spectra of GaN nanocolumns are shown in Figure 13. The PL peak of all GaN nanocolumns is located at about 364 nm, which corresponds to the edge absorption of GaN. The peak of the PL intensity becomes higher with the decrease in diameter due to the generation of more crystal defects. These defects become non-radiative recombination centers that trap the photo-generated carriers and lower the radiative recombination. Moreover, insufficient band bending causes the transport of photo-generated carriers with low efficiency, which will be discussed later.

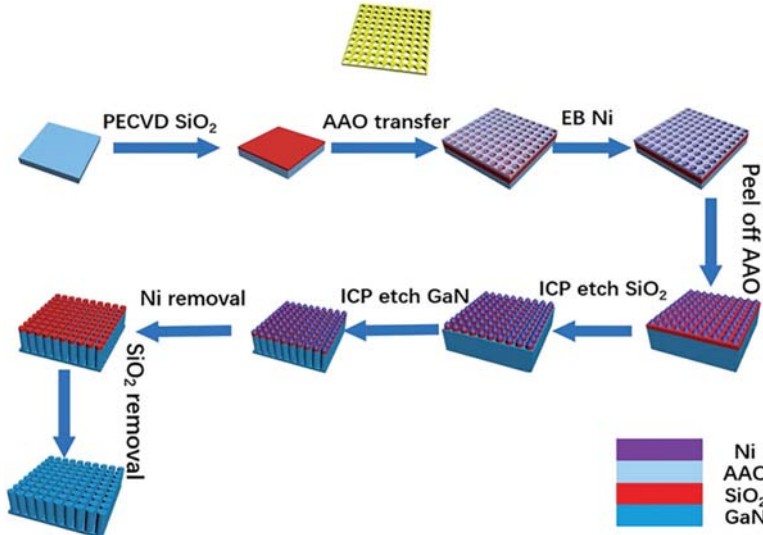

**Figure 11.** Illustration of the fabrication of GaN nanocolumns using an AAO mask as the pattern template. Figure reprinted from [84] under the Creative Common Attribution 3.0 Unported License.

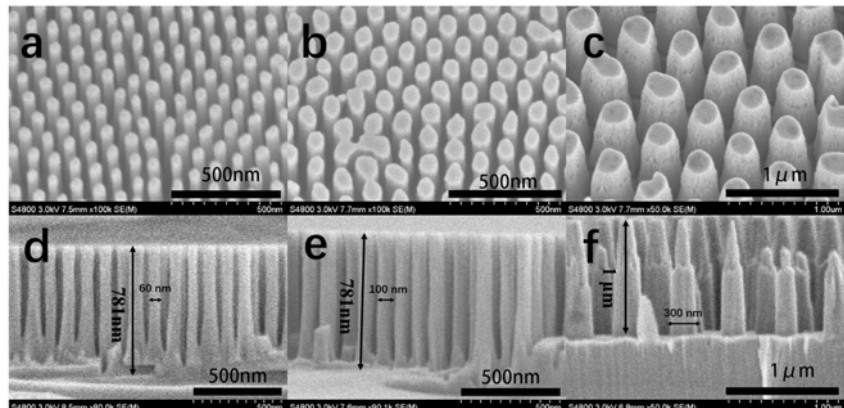

**Figure 12.** Top-tilted SEM images (tilted at 30° angle) (**a**–**c**) and cross-sectional SEM images (**d**–**f**) of GaN nanocolumns with diameters of 60, 100, and 300 nm. Figure reprinted from [84] under the Creative Common Attribution 3.0 Unported License.

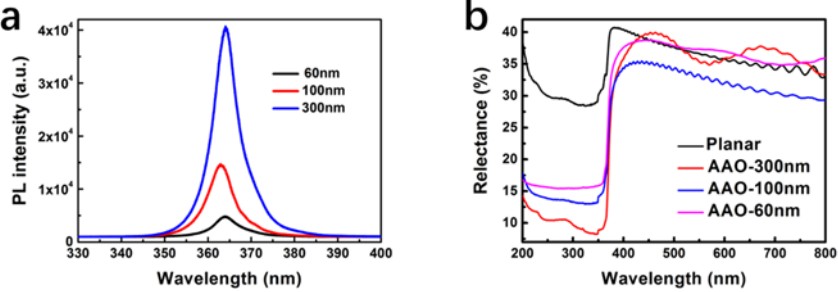

**Figure 13.** Room-temperature PL (**a**) and reflectance spectra (**b**) of the GaN nanowires with diameters of 60, 100, 300 nm and planar GaN. Figure reprinted from [84] under the Creative Common Attribution 3.0 Unported License.

The reflectance of all samples increases in ascending order from planar GaN to GaN nanocolumns with diameters of 60, 100, and 300 nm. The higher reflectance implies lower light absorption, which illustrates that GaN nanocolumns have stronger light absorption compared with planar GaN. As the diameters increase, the light absorption increases. In

the reflection spectrum, the absorption of the GaN nanocolumns redshifts about 45 nm, which is due to the band bending result from the surface state.

For the n-type semiconductor, the surface state bends the energy band upwards, resulting in the formation of a space charge region where electrons accumulate at the bottom of the energy band and holes accumulate at the top, as illustrated in Figure 14a. As a result, photo-generated carriers separate promptly to avoid mutual recombination. The fast separation of carriers reduces their recombination rate and lowers the reflection rate. The diameters of the nanocolumns have an important influence on the space charge region, as depicted in Figure 14.

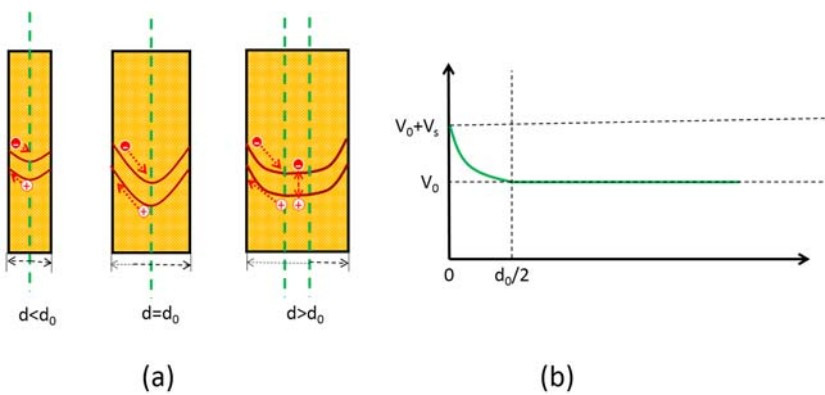

**Figure 14.** (**a**) Schematic diagram of the Fermi level at the GaN nanowires surface [84]. (**b**) Schematic of calculation of maximum width of GaN nanocolumns. Figure reprinted from [84] under the Creative Common Attribution 3.0 Unported License.

There should be a critical diameter $d_0$ which achieves the optimum separation efficiency for carriers. When the diameter is smaller than $d_0$, the energy band cannot bend to its maximum, resulting in an internal electric field that does not reach its strongest value. Consequently, photo-generated carriers cannot gain enough acceleration and may have a higher chance of recombination with each other. In contrast, when the diameter is equivalent to $d_0$, the energy band obtains its maximum curvature, and the internal electric field approaches its strongest value. The electric field then causes the separation of photo-generated carriers to achieve optimal efficiency. However, when the diameter is larger than $d_0$, a flat band will appear in the region except the space charge region, causing photo-generated carriers to recombine with each other on the flat band. The value of $d_0$ can be calculated using the following equation:

$$\frac{\mathrm{d}^2\phi(\mathrm{r})}{dr2} = -\frac{eN(r)}{\varepsilon_s\varepsilon_0} \tag{4}$$

where $\varphi(r)$ represents the surface energy, $e$ is the elementary charge and $N(r)$ is the carrier density at position $r$, and $\varepsilon_r$ and $\varepsilon_0$ are the electric constant and vacuum permittivity of GaN, respectively.

According to the calculation, a diameter of GaN nanocolumns of about 140 nm can attain optimal separation efficiency for carriers. The water splitting performances of GaN nanocolumns are illustrated in Figure 15, demonstrating that they outperform planar GaN. This improvement is due to the advantages of nanostructures, including a larger surface-to-volume ratio, shorter transport distance of carriers, and enhanced light absorption, as discussed previously. The water splitting performances of GaN nanocolumns decrease from the largest to smallest diameters (300, 100, 60 nm), corresponding to the reflection spectrum in reverse order. This result is consistent with the earlier analysis of the diameter of nanocolumns.

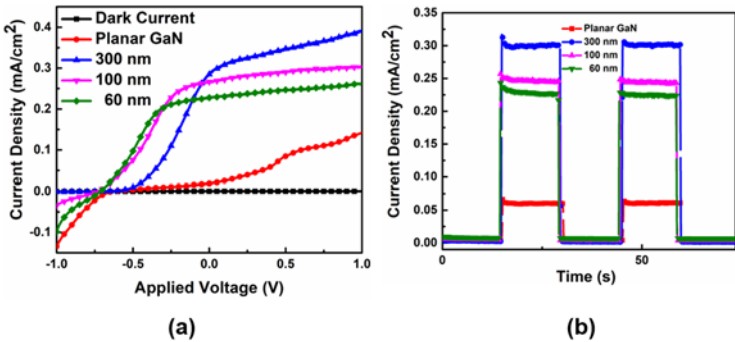

**Figure 15.** J-V curves (**a**) and I-t curves (**b**) of GaN nanocolumns with diameters of 60, 100, and 300 nm as well as planar GaN. Figure reprinted from [84] under the Creative Common Attribution 3.0 Unported License.

Annealing Ni thin film as an ICP etching mask is another cost-effective method for fabricating GaN nanocolumns [74]. The fabrication process scheme is illustrated in Figure 16. Depositing Ni films with different thicknesses of Ni film results in diverse sizes of Ni nanoparticles. GaN nanocolumns with diameters of 170, 220, and 250 nm were fabricated by depositing Ni films with 7, 11, and 15 nm thicknesses, respectively (Figure 17). The diameter of the GaN nanocolumns increases with the increase in Ni deposition thickness. The patterns transferred from the Ni particles are random within a certain range at the center of a specific value, as shown in the schematic diagram.

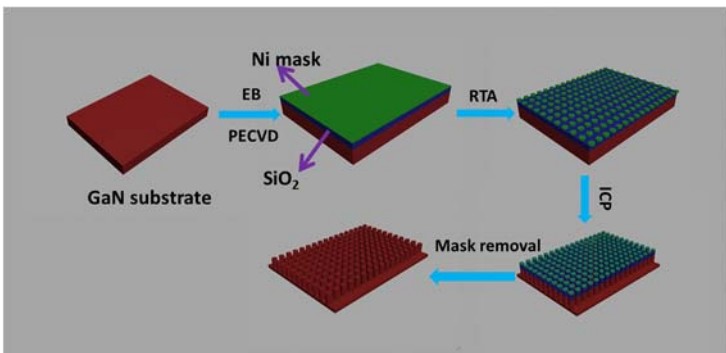

**Figure 16.** Illustration of the scheme for the fabrication of GaN nanocolumns using the top-down method [74]. Reprinted with permission from [74] 2018, Elsevier.

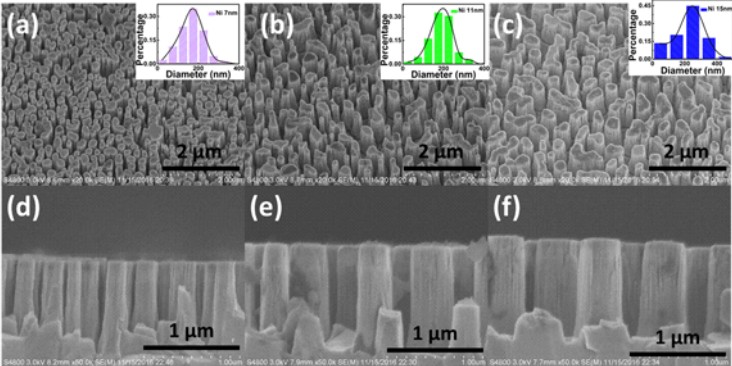

**Figure 17.** Top-view SEM images [74] (images taken at tilted 30° angle) of GaN nanocolumns with increasing Ni mask thicknesses of 7 (**a**), 11 (**b**), and 15 (**c**) nm. and cross-sectional SEM images of GaN nanocolumns with increasing Ni mask thicknesses of 7 (**d**), 11 (**e**), and 15 (**f**) nm. Reprinted with permission from [74] 2018, Elsevier.

The water splitting performances of the GaN nanocolumns are displayed in Figure 18. The PEC performances of the GaN nanocolumns are better than the planar GaN which is consistent with the results of the GaN nanocolumns fabricated by AAO transfer. This improvement is due to the advantages of higher light absorption, faster carrier transportation, and more reactive sites with the electrolyte. It is noteworthy that the photocurrent increases with the decrease in GaN nanocolumn diameter, which contradicts the trend observed in the GaN nanocolumns fabricated with AAO transfer. As concluded in the case of AAO transfer, the diameter of the GaN nanocolumns has an immerse influence on their water splitting performance. For the 2 experiments involving GaN, the best water splitting performance was at a diameter of 140 nm. When the diameter is smaller or larger than 140 nm, the photocatalytic property drops due to insufficient acceleration or recombination on a flat band.

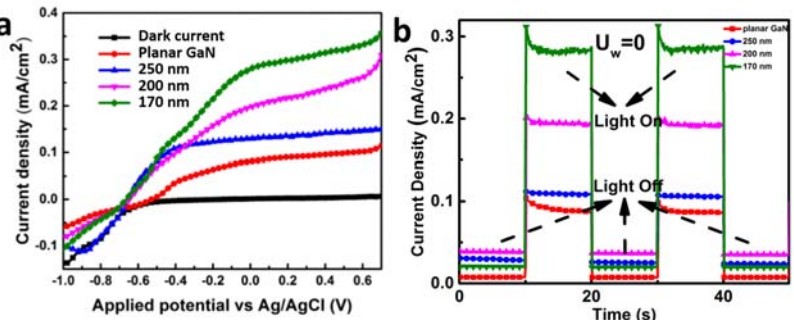

**Figure 18.** (**a**) J-V curves and (**b**) I-t curves of GaN nanocolumns as well as planar GaN [74]. Reprinted with permission from [74] 2018, Elsevier.

## 5. Prospects for the Fabrication of GaN Nanocolumns

This article has introduced various technologies for fabricating GaN nanostructures. However, there is still much room for improvement in this field. The bandgap of GaN-based ternary alloys can be adjusted by adding proper In composition, thereby extending light absorption to the full solar range. Incorporating cocatalysts can lower the reaction activation energy and improve the water splitting efficiency. Constructing heterojunction-based GaN nanostructures can enhance the water splitting performance by facilitating multiple light absorption and carrier acceleration due to the internal electric field. Improving the water splitting performance of GaN is still crucial for the future use of clean energy. The fabrication of GaN nanostructures merely indicates a direction for the development of GaN materials in water splitting applications. More research efforts are needed to promote the progress of GaN in this area.

## 6. Conclusions

GaN is one of the most promising materials that has attracted the attention of many scientists. This article mainly discusses the cost-effective methods for fabricating GaN nanostructures, including electrochemical etching of nanoporous GaN and top-down methods for fabricating GaN nanocolumns. The influence of voltage on the size of nanopores and formation mechanics are emphasized in the electrochemical etching of lateral nanoporous GaN. Moreover, the fabrication of vertical nanoporous GaN and composite nanoporous GaN are also introduced for further improvement of water splitting performance. This paper also presents two cost-effective schemes for an AAO membrane and annealing of Ni thin film as an etching mask. The theoretical analysis of the relation between the diameters of GaN nanocolumns and the inner electric field shows that the best diameter for GaN nanocolumns is about 140 nm. Water splitting measurements demonstrate that GaN nanostructures exhibit excellent properties compared with planar GaN due to their larger surface areas, enhanced light absorption, and facile reaction conditions. Based on the fabrication of GaN nanostructures, recommendations have been made for further photocat-

alytic improvement of GaN, including narrower bandgap InGaN alloys, heterojunctions, and cocatalysts.

**Author Contributions:** Conceptualization, L.Z. and C.Y.; methodology, X.L.; validation, L.Z., C.Y. and T.L.; formal analysis, X.X. and C.Y.; investigation, L.Z., C.Y., X.X. and X.L.; resources, T.L.; data curation, T.L.; writing—original draft preparation, X.X. and T.L.; writing—review and editing, L.Z.; visualization, C.Y., X.L. and X.X.; supervision, L.Z. and T.L. All authors have read and agreed to the published version of the manuscript.

**Funding:** This work was supported by the National Natural Science Foundation of China (Grant No. 61774148 and 11974343) and the National Key Research and Development Program of China (No. 2017YFB0403601 and 2017YFB0403602).

**Data Availability Statement:** Data sharing is not applicable to this article as no new data were created or analyzed in this study.

**Conflicts of Interest:** The authors declare no conflict of interest.

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
