# Peer review of "The Fabrication of GaN Nanostructures Using Cost-Effective Methods for Application in Water Splitting"

_crystals, doi:10.3390/cryst13060873_

Round 1

Reviewer 1 Report

In the manuscript, the authors have attempted to provide an overview of the methods used to fabricate three types of GaN nanostructures: lateral nanopores, vertical nanopores and nanopillars for water spitting applications. They limited themselves to a single, cost-effective method: electrochemical anodisation. At the end, they gave a brief overview with general remarks on the prospects for GaN nanocolumns fabrication.

They failed to give an overview of the water splitting process and the results of the water splitting process with GaN nanostructures. They do not clearly state and discuss what material properties are important and necessary for application in the water splitting process in general. You have not explained what it means to produce GaN nanostructures in a cost-efficient way. What are the criteria for this?

The aim of the manuscript is not well described. In the introduction, the authors write: "By the reason of above, we will emphatically introduce the fabrication methods of GaN nanopores and nanocolumns by chemical etching. We will mainly investigate the influence of different parameters on GaN morphology and water splitting performance,such as the applied voltage, diameters of nanocolumns, diameters of nanopores and so on.". However, they have only provided an overview of previously published work on GaN nanostructure fabrication.

Searching literature much informative review manucripts on the fabrication and application of GaN for water spitting can be found. For example: Gao F, Liu Q, Shi J, et al. (2021) Recent Progress in Gallium Nitride for Photoelectrochemical Water Splitting. Nanowires - Recent Progress. IntechOpen. DOI: 10.5772/intechopen.92848.

Moreover, the content is not clear and expressed in good English. The English should be improved. Many unconventional and incomprehensible terms are used, as well as unclear sentences.For example:

    • third semiconductor? Correct term is third-generation semiconductor

    • optical-electro properties?

    • “Besides nanopores, nanocolumns is another principal method for fabricating GaN nanostructures[46,58-61].”

    • chemical etching or electrochemical etching

    • “ The schematic process diagram for the laterally porous GaN.”

Terminology used in manuscript should be standardised. For example: 

    • chemical etching or electrochemical etching

The figurer captions are not precise and complete. For example, the caption of figure 6 is incomplete and incorrect. Fig. 6a does not show an etching current. Description for Fig. 6b,c,d,e is missing. Similar for Fig. 5 and Fig. 13, etc.

There are many more minor comment and suggestions. For example: 

    • Figure 4: The names of the specimens are not clearly described in the text or in the caption. 

    • The source reference is missing for some figures, e.g. Fig. 14 

    • Several abbreviations are used without prior definition. For example FDTD. The manuscript should be checked for this and edited accordingly.

The English should be improved. Many unconventional and incomprehensible terms are used, as well as unclear sentences. For example:

    • third semiconductor? Correct should be third-generation semiconductor

    • optical-electro properties?

    • “bination[53,57].

    • “Besides nanopores, nanocolumns is another principal method for fabricating GaN nanostructures[46,58-61].”

    • chemical etching or electrochemical etching

    • “ The schematic process diagram for the laterally porous GaN.”

Author Response

Thank you for your precious suggestions. I have added the overview of water splitting process and material properties which are important and necessary for water splitting application in paragraph 3 page 1 and paragraph 2 page 2. “The principle of water splitting reaction mainly involves three stages, which are presented in Figure 1 [2-4]. First, the generation of electron-hole pairs on the photoanode. When the energy of incident photons is greater than the bandgap of the photoanode, it excites many electron-hole pairs on its energy band. Second, the transfer of photo-generated electron-hole pairs to the surface of the photoanode. Due to diffusion or polarization effects, the electron-hole pairs migrate toward the surface of the photoanode. During this process, some photo-generated carriers are trapped in crystal defects or recombine with each other, resulting in energy loss. Thus, improving the crystal quality and designing a fast separation mechanism for electron-hole pairs can significantly enhance the water splitting performance. Stage 3 involve the oxidation and reduction of OH- and H+. If the electrons are more negative than 0 v vs NHE (normal hydrogen electrode), they will reduce H+ into H2. If the holes are more positive than 1.23 v vs NHE, they will oxidize OH- into O2. The equation for water splitting reaction is shown below:

H2O → 1/2O2 + H2;    ΔG = +237kJ/mol, E = 1.23 eV   (1-1)

                 2H+ + 2e- → H2;    (ERe = 0 eV)                    (1-2)

       2H2O → 4H+ + 4e- +O2;    (EOx = 1.23 eV)             (1-3)

Figure 1. The schematic diagram of water splitting process.

Therefore, the bandgap of the photoanode must be larger than 1.23 eV, with the potential of electrons on the conduction band more negative than 0 V and holes on the valence band more positive than 1.23 V vs NHE. For fast separation of photo-generated carriers, the photoanode should be designed with a strong internal electric field to accelerate the transport of electron-hole pairs, a large surface-to-volume ratio to absorb more photons and provide more reaction sites, and a narrow bandgap to increase the solar absorption range.”

I have explained what it means to produce GaN nanostructures in a cost-efficient way in paragraph 2 and 3 page 3. “The fabrication methods of GaN nanostructures mainly include MOCVD, SAG, VLS, MBE, HVPE, electrochemical etching, and ICP etching [46,58-78]. Among them, MOCVD and SAG cannot fabricate homogeneous morphology and stable properties of GaN nanostructures. MBE can grow materials with atom precision but is unsuitable for large-scale production due to its slow growth rate and limited growth area [62-64]. The HVPE method has difficulty in doping, which results in the inability to grow GaN-based materials with high carrier mobility.

The epitaxy technology for GaN wafers has developed from prototype to commercial application over about thirty years. The epitaxy of GaN with a large size and high quality has been matured for mass production for a long time. Based on mass-produced GaN wafers, chemical etching can be adopted to fabricate GaN nanostructures with stable properties on a large scale. Therefore, the GaN nanostructures fabricated by chemical etching are currently the most cost-effective way for quantitative production.”

This review article primarily focuses on cost-effective methods for fabricating GaN nanostructures, and includes extensive coverage of our group's work in this area, which is cited throughout the article. Additionally, we have included important contributions from other research groups in this field to provide a comprehensive overview of the current state-of-the-art techniques. By highlighting both our own findings and those of other researchers, this review offers a more complete understanding of the advantages and limitations of different approaches for producing GaN nanostructures in a cost-effective manner. 

Reviewer 2 Report

The authors discuss the different possible cost-effective ways to fabricate GaN nanostructures for water splitting. while they discuss the different techniques, there is no discussion regarding the cost efficiency of the processes. Since the paper is focused on "cost-effective", a discussion on the cost-effectiveness of the processes, compared to other ones need to be presented. Also the total efficiency of the whole process need to be discussed and energy output per dollar need to be assessed. This way, the review will offer a useful insight in the future prospect of these techniques.

Aside from this major comment, there is no comment on the technical aspects of the manuscript.

Author Response

Thank you for your precious suggestions about our paper. We have incorporated your valuable feedback into our paper, particularly in regards to discussing the cost efficiency of chemical etching. In paragraphs 3 and 4 on page 3, we have provided a comprehensive summary of the fabrication methods for GaN nanostructures and have analyzed the reasons why chemical etching is a cost-effective option. “The fabrication methods of GaN nanostructures mainly include MOCVD, SAG, VLS, MBE, HVPE, electrochemical etching, and ICP etching [46,58-78]. Among them, MOCVD and SAG cannot fabricate homogeneous morphology and stable properties of GaN nanostructures. MBE can grow materials with atom precision but is unsuitable for large-scale production due to its slow growth rate and limited growth area [62-64]. The HVPE method has difficulty in doping, which results in the inability to grow GaN-based materials with high carrier mobility.

The epitaxy technology for GaN wafers has developed from prototype to commercial application over about thirty years. The epitaxy of GaN with a large size and high quality has been matured for mass production for a long time. Based on mass-produced GaN wafers, chemical etching can be adopted to fabricate GaN nanostructures with stable properties on a large scale. Therefore, the GaN nanostructures fabricated by chemical etching are currently the most cost-effective way for quantitative production.”

While there is currently no standard criterion for assessing energy output per dollar for applications of GaN nanostructures in research, when it comes to mass production or large-scale fabrication, chemical etching has numerous advantages and offers a highly cost-effective method.

Reviewer 3 Report

The research is quite interesting for readers. The article discusses the possible application of GaN nanostructures for water splitting, which is currently an extremely relevant topic. However, the article needs significant revisions before publication.

1) The introduction does not clearly indicate the novelty of this work compared to existing research. The authors are requested to add to the introduction and clearly highlight the novelty of this work.

2) The text lacks coherence and logical flow at times. For instance, the discussion on GaN nanocolumns and nanopores should be more focused and directly linked to the topic of water splitting. Additionally, the discussion on heterojunctions and co-catalysts should be more detailed, explaining their roles in improving watersplitting performance.

3) The most significant issue requiring detailed revision in this work is the illustrations. The fonts used on the figures are too small, and sometimes the figures themselves are too small, making the work partially unreadable.

In Figure 2, the SEM is not visible, and the statistics graphs (2(b)) are entirely illegible.

In Figure 5, the values on the right-hand scale are not visible.

Other figures could be reorganized for better readability (in particular, Figure 6 could occupy the entire width of the field).

I recommend that the authors systematically go through all the figures and evaluate the readability of each of them

I don't see any serious issues with the language, but English is not my native language.

Author Response

  • The introduction does not clearly indicate the novelty of this work compared to existing research. The authors are requested to add to the introduction and clearly highlight the novelty of this work.

Answer: Thank you for your precious suggestions. In paragraphs 2-4 on page 3, we have provided an extensive discussion of various preparation methods for GaN nanostructures, with a focus on the advantages of chemical etching. Our research draws from the latest studies published in recent years, and we believe this comprehensive overview will provide valuable insights for researchers in the field. Particularly in paragraph 2, 3, 4, we highlight the benefits of chemical etching, emphasizing its strengths as a low-cost, high-throughput method for fabricating GaN nanostructures on a large scale.

2) The text lacks coherence and logical flow at times. For instance, the discussion on GaN nanocolumns and nanopores should be more focused and directly linked to the topic of water splitting. Additionally, the discussion on heterojunctions and co-catalysts should be more detailed, explaining their roles in improving watersplitting performance.

Answer: Thank you for your honorable advise. This research article primarily focuses on the fabrication methods of GaN nanostructures and their application in water splitting. Through various sections in the paper—specifically, paragraph 2 on page 5, paragraph 2 on page 6, paragraph 2 on page 7, paragraphs 1 and 3 on page 8, and page 9—we have highlighted the advantages that GaN nanostructures can offer in improving water splitting efficiency. These benefits range from increased surface-to-volume ratio and enhanced light absorption capabilities to reduced carrier-electrolyte distances and improved gas ventilation to surface states. However, as our article is mainly focused on summarizing cost-effective methods for fabricating GaN nanostructures, we will explore other improvements, such as incorporating heterojunctions and cocatalysts, in a subsequent work.

3) The most significant issue requiring detailed revision in this work is the illustrations. The fonts used on the figures are too small, and sometimes the figures themselves are too small, making the work partially unreadable.

Answer: Thank you for your precious suggestion. I have checked all the figures in our paper and make the necessary modification. And I have found that due to the difference of computer resolution, the fonts used on the figures maybe ambiguous. Because I used one computer to revise my paper and found the same problem. I have make a copy of our paper in PDF version for your check. If there still are some prob

Round 2

Reviewer 1 Report

The authors have answered to all questions and comments and modified the manuscript according to the review comments.

Reviewer 3 Report

The article has been significantly improved and can be published in its current form